# DISCRIMINATIVE OUT-OF-DISTRIBUTION DETECTION FOR SEMANTIC SEGMENTATION

## ABSTRACT

Most classification and segmentation datasets assume a closed-world scenario in which predictions are expressed as distribution over a predetermined set of visual classes. However, such assumption implies unavoidable and often unnoticeable failures in presence of out-of-distribution (OOD) input. These failures are bound to happen in most real-life applications since current visual ontologies are far from being comprehensive. We propose to address this issue by discriminative detection of OOD pixels in input data. Different from recent approaches, we avoid to bring any decisions by only observing the training dataset of the primary model trained to solve the desired computer vision task. Instead, we train a dedicated OOD model which discriminates the primary training set from a much larger "background" dataset which approximates the variety of the visual world. We perform our experiments on high resolution natural images in a dense prediction setup. We use several road driving datasets as our training distribution, while we approximate the background distribution with the ILSVRC dataset. We evaluate our approach on WildDash test, which is currently the only public test dataset that includes out-of-distribution images. The obtained results show that the proposed approach succeeds to identify out-of-distribution pixels while outperforming previous work by a wide margin.

## 1 INTRODUCTION

Development of deep convolutional models has resulted in tremendous advances of visual recognition. Recent semantic segmentation systems surpass 80% mIoU (Chen et al., 2017) on demanding natural datasets such as Pascal VOC 2012 (Everingham et al., 2015) or Cityscapes (Cordts et al., 2015). Such performance level suggests a clear application potential in exciting areas such as road safety assessment or autonomous driving. Unfortunately, most existing semantic segmentation datasets assume closed-world evaluation (Scheirer et al., 2013), which means that they require predictions over a predetermined set of visual classes. Closed-world datasets are very useful for promoting research, however they are poor proxies for real-life operation even in a very restricted scenario such as road driving. In fact, one can easily imagine many real-life driving scenes which give rise to image regions that can not be recognized by learning on the Cityscapes ontology. Some of those regions may be projected from objects which are foreign to Cityscapes (e.g. road works, water, animals). Other may appear unrelated to Cityscapes due to particular configurations being absent from the training dataset (e.g. pedestrians lying on the ground, crashed cars, fallen trees). Finally, some regions may be poorly classified due to different environmental conditions, acquisition setup, or geographical location (Tsai et al., 2018).

The simplest way to approach unrecognizable data is to improve datasets. For instance, the Vistas dataset (Neuhold et al., 2017) proposes a richer ontology and addresses more factors of variation than Cityscapes. However, training on Vistas requires considerable computational resources while still being unable to account for the full variety of the recent WildDash dataset (Zendel et al., 2018), as we show in experiments. Another way to approach this problem would be to design strategies for knowledge transfer between the training dataset and the test images (Tsai et al., 2018). However, this is unsatisfactory for many real world applications where the same model should be directly applicable to a variety of environments.

These examples emphasize the need to quantify model prediction uncertainty, especially if we wish to achieve reliable deployment in the real world. Uncertainty can be divided into two categories (Kendall & Gal, 2017). Aleatoric uncertainty is caused by limitations of the model which cannot be reduced by supplying additional training data. For example, the quality of segmentation models on distant and small objects depends on the resolution on which inference is performed. On the other hand, epistemic uncertainty arises when the trained model is unable to bring the desired prediction given particular training dataset. In other words, it occurs when the model receives the kind of data which was not seen during training. Epistemic uncertainty is therefore strongly related to the probability that the model operates on an out-of-distribution sample.

Recent work in image-wide out-of-distribution detection (Kendall & Gal, 2017; Hendrycks & Gimpel, 2017; Liang et al., 2018) evaluates the prediction uncertainty by analyzing the model output. We find that these approaches perform poorly in dense prediction tasks due to prominence of aleatoric uncertainty. This means that total uncertainty can be high even on in-distribution pixels (e.g. on pixels at semantic borders, or very distant objects).

A different approach attempts to detect out-of-distribution samples with GAN discriminators, whereby the GAN generator is used as a proxy for the out-of-distribution class (Lee et al., 2018; Sabokrou et al., 2018). However, these approaches do not scale easily to dense prediction in high resolution images due to high computational complexity and large memory requirements.

Therefore, in this paper we propose to detect out-of-distribution samples on the pixel level by a dedicated "OOD" model which complements the "primary" model trained for a specific vision task. We formulate the OOD model as dense binary classification between the training dataset and a much larger "background" dataset. The proposed formulation requires less computational resources than approaches with GAN-generated backgrounds, and is insensitive to aleatoric uncertainty related to semantic segmentation.

## 2 RELATED WORK

Detection of out-of-distribution (OOD) examples (together with related fields of anomaly and novelty detection) have received a lot of attention in recent literature. Many approaches try to estimate uncertainty by analyzing entropy of the predictions. The simplest approach is to express the prediction confidence as the probability of the winning class or, equivalently, the maximum softmax (max-softmax) activation (Hendrycks & Gimpel, 2017). The resulting approach achieves useful results in image classification context, although max-softmax must be recalibrated (Guo et al., 2017) before being interpreted as $P(\text{inlier}|\mathbf{x})$. This result has been improved upon by the approach known as ODIN (Liang et al., 2018) which proposes to pre-process input images with a well-tempered anti-adversarial perturbation with respect to the winning class and increase the softmax temperature.

Another line of research characterizes uncertainty by a separate head of the primary model which learns either prediction uncertainty (Kendall & Gal, 2017; Lakshminarayanan et al., 2017) or confidence (DeVries & Taylor, 2018). The predicted variance (or confidence) is further employed to discount the data loss while at the same time incurring a small penalty proportional to the uncertainty (or inversely proportional to confidence). This way, the model is encouraged to learn to recognize hard examples if they are present in the training dataset. Unfortunately, such modelling is able to detect only aleatoric uncertainty (Kendall & Gal, 2017), since the data which would allow us to learn epistemic uncertainty is absent by definition.

A principled information-theoretic approach for calculating epistemic uncertainty has been proposed by Smith & Gal (2018). They express epistemic uncertainty as mutual information between the model parameters and the particular prediction. Intuitively, if our knowledge about the parameters increases a lot when the ground truth prediction becomes known, then the corresponding sample is likely to be out of distribution. The sought mutual information is quantified as a difference between the total prediction entropy and the marginalized prediction entropy over the parameter distribution. Both entropies are easily calculated with MC dropout. Unfortunately, our experiments along these lines resulted in poor OOD detection accuracy. This may be caused by MC dropout being an insufficiently accurate approximation of model sampling according to the parameter distribution, however, further work would be required in order to produce a more definitive answer.

Lakshminarayanan et al. (2017) show that uncertainty can be more accurately recovered by replacing MC dropout with an ensemble of several independently trained models. However, for this to be done, many models would need to be in GPU memory during evaluation. Explicit ensembles are therefore not suited for systems which have ambition to perform dense prediction in real time.

A principled algorithm for recognizing OOD samples would fit a generative model $P_\mathcal{D}(\mathbf{x}|\theta)$ to the training dataset $\mathcal{D} = \{\mathbf{x}_i\}$. Such model would learn to evaluate the probability distribution of the training dataset at the given sample, which can be viewed as epistemic uncertainty. Unfortunately, evaluating probability density function for high-dimensional data is very hard. Instead Lee et al. (2018); Sabokrou et al. (2018) formulate OOD detection as binary classification where OOD samples are produced by a GAN generator (Goodfellow et al., 2014). Sabokrou et al. (2018) use an autoencoder in place of the generator, which is trained to denoise images as well as to trick the discriminator. During evaluation the autoencoder enhances inliers while distorting the outliers, making the two more separable. Lee et al. (2018) train a GAN to generate images on the borders of the distribution. In both of these approaches the discriminator can be used for OOD detection. Unfortunately, these approaches have been shown to work on low dimensional samples and simple image datasets. They do not scale well to pixel level prediction of a highly complex domain such as traffic scenes, especially if we take into account memory and time restrictions of real world applications.

While Lee et al. (2018) try to generate outliers, it may be simpler to just use a foreign dataset during training. Kreso et al. (2018) show that out-of-distribution samples for traffic images can be detected by joint segmentation training on Cityscapes (road driving) and ScanNet (indoor). Out-of-distribution sample is signaled whenever the prediction favours a class which is foreign to the evaluation dataset. For instance, if a bathtub region is detected in an image acquired from an intelligent vehicle, then the pixels of that region are labeled as OOD.

## 3 THE PROPOSED DISCRIMINATIVE OOD DETECTION APPROACH

We formulate out-of-distribution pixel detection as dense binary classification into two classes: OOD (out-of-distribution) and ID (in-distribution). As in (Lee et al., 2018; Sabokrou et al., 2018), we take ID training pixels from the semantic segmentation dataset used to train the primary semantic segmentation model (e.g. Vistas). Subsequently, we propose to take the OOD training pixels from a foreign dataset which approximates the entire diversity of the visual world.

Since none of existing public datasets is able to satisfy our requirements, in this paper we propose to use the ILSVRC dataset (Russakovsky et al., 2015) as the next-best solution for this purpose (cf. Figure 1). We think that this approximation is reasonable since the ILSVRC dataset has been successfully used in many knowledge transfer experiments (Oquab et al., 2014; Garcia-Gasulla et al., 2018). Furthermore, ILSVRC images contain a variety of content around the 1000 official classes.

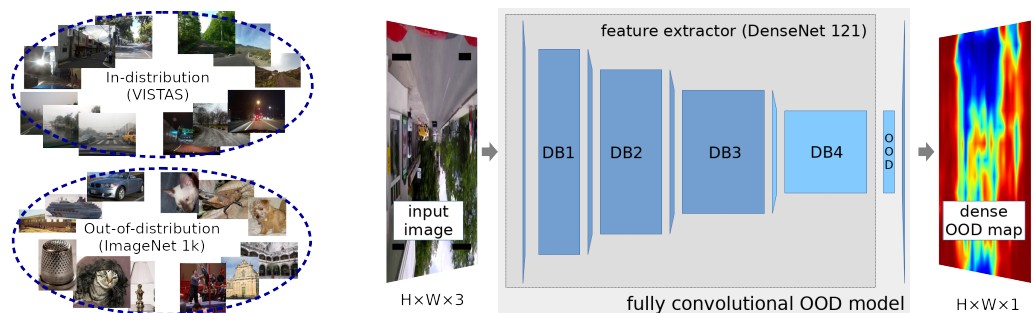

Figure 1: The proposed discriminative OOD detection approach for semantic segmentation. Training in-distribution (ID) images are taken from a diverse road driving dataset (e.g. VISTAS). Training out-of-distribution (OOD) images are taken from ILSVRC, which is much more diverse than ID images. Our model is composed of a feature extractor pretrained on ImageNet and the OOD detection head. The model transforms colour images H×W into H/32×W/32 logits. The cross-entropy loss is applied to bilinearly upsampled logits at input resolution. We train only the OOD head, DB4 and the preceeding transition layer (light blue), while we freeze the remaining layers (dark blue).

We define the learning algorithm in a way to avoid ILSVRC images hijacking content from the primary training dataset. We observe two distinct problems: i) ILSVRC classes which overlap classes from the primary dataset, e.g. a car in an ILSVRC image labeled as car, and ii) primary dataset classes in the ILSVRC image background, e.g. a person playing a banjo in an ILSVRC image labeled as banjo. Currently, we address both problems by ensuring that the model is trained on in-distribution (ID) pixels as often as on OOD ones. Due to diversity of the OOD class (ILSVRC) such training results in a bias towards the ID dataset. For example, there are around 10 car-like classes in ILSVRC; therefore, cars occur in only around 1/100 ILSVRC images. On the other hand, almost every image from any road driving dataset is likely to contain at least one car. Hence, the proposed approach ensures that the model is more likely to classify car pixels as inliers (Cityscapes) rather than as outliers (ILSVRC). We consider several improvements to this approach in the appendix.

The proposed OOD model is discriminative and fully convolutional, and we train it with the dense cross-entropy loss. During evaluation, we apply the trained model to test images in order to obtain dense logit maps for the two classes. These logits reflect the epistemic uncertainty in each individual pixel. OOD detection is finally performed by thresholding the probability of the inlier class.

The proposed OOD model works well in tandem with ILSVRC because it can be warmed-up with parameters pre-trained on the same dataset. This considerably speeds-up the training since the optimization starts from a model which is already able to extract useful features, and only needs to learn to distinguish between ILSVRC and the in-distribution dataset. This approach can share the common feature extractor with the primary semantic segmentation model, which results in a smaller OOD-detection overhead than in GAN-based approaches (Lee et al., 2018; Sabokrou et al., 2018).

## 4 EXPERIMENTS

Our experiments explore the accuracy of dense OOD detection in realistic natural images from several driving datasets. We train our models on different datasets and discuss the obtained performance on WildDash test and other datasets. Experiments compare our approach to several previous approaches from the literature, which we adapt for dense prediction as explained next.

### 4.1 ADAPTING IMAGE-WIDE APPROACHES FOR DENSE PREDICTION

We consider three previous approaches which were originally developed for image-wide OOD detection and adapt them for dense prediction. For the max-softmax approach (Hendrycks & Gimpel, 2017), the task is trivial: it is enough to independently assess the prediction of the primary model in each image pixel.

For the ODIN approach (Liang et al., 2018), we first perturb the input image in the direction which increases the probability of the winning class in each individual pixel (according to the primary model). Consequently, we apply the softmax temperature and consider the max-softmax response as above. The perturbation magnitude $\epsilon$ and softmax temperature $T$ are hyperparameters that need to be validated. Note that dense ODIN perturbation implies complex pixel interactions which are absent in the image-wide case. Despite hyperparameter tuning, this achieved only a modest improvement over the max-softmax approach so we do not present it in the tables.

For trained confidence (DeVries & Taylor, 2018), we introduce a separate convolutional head to the primary model. The resulting confidence predictions diminish the loss of wrong prediction in the corresponding pixel while incurring a direct loss multiplied with a small constant.

### 4.2 DATASETS

Cityscapes is a widely used dataset (Cordts et al., 2015) containing densely annotated images from the driver perspective acquired during rides through different German cities. It is divided into training (2975 images), validation (500 images) and test subsets (1525 images). Vistas (Neuhold et al., 2017) is larger and more diverse than Cityscapes. It contains much more diversity with respect to locations, time of day, weather, and cameras. There are 18 000 train and 2 000 validation images.

The WildDash dataset (Zendel et al., 2018) provides a benchmark for semantic segmentation and instance segmentation. It focuses on providing performance-decreasing images. These images are

challenging due to conditions and unusual locations in which they were taken or because they contain various distortions. There are 70 validation and 156 test images. The test set contains 15 images which are marked as negatives. All pixels in these images are considered out-of-distribution in the context of semantic segmentation on road driving datasets. These images contain noise, indoor images, and five artificially altered inlier images (see Figure 3). WildDash is compatible with Cityscapes labeling policy (Cordts et al., 2015), but it also considers performance on negative images. Pixels in negative images are considered to be correctly classified if they are assigned a correct Cityscapes label (e.g. people in an indoor scene) or if they are assigned a void label (which means that they are detected as OOD samples). The official WildDash web site suggest that OOD pixels could be detected by thresholding the max-softmax value.

The ILSVRC dataset (Russakovsky et al., 2015) includes a selection of 1000 ImageNet (Deng et al., 2009) classes. It contains 1 281 167 images with image-wide annotations and 544 546 images annotated with the bounding box of the object which defines the class. The training split contains over one million images, while the validation split contains 50 000 images.

The Pascal VOC 2007 dataset contains 9 963 training and validation images with image-wide annotations into 20 classes. Pixel-level semantic segmentation groundtruth is available for 632 images.

## 4.3 Model details

We experiment with three different models: i) the primary model for semantic segmentation, ii) the augmented primary model with confidence (DeVries & Taylor, 2018), and iii) the proposed discriminative model. All three models are based on the DenseNet-121 architecture (Huang et al., 2017) (we assume the BC variant throughout the paper). DenseNet-121 contains 120 convolutional layers which can be considered as a feature extractor. These layers are organized into 4 dense blocks (DB_1 to DB_4) and 3 transition layers (T_1 to T_3) inbetween. In all of our models, the feature extractor is initialized with parameters pretrained on ILSVRC.

We build our primary model by concatenating the upsampled output of DB_4 with the output of DB_2. This concatenation is routed to the spatial pyramid pooling layer (SPP) (He et al., 2014) which is followed by a BN-ReLU-Conv block (batch normalization, ReLU activation, 3x3 convolution) which outputs 19 feature maps with logits. The logits are normalized with softmax and then fed to the usual cross-entropy loss.

We augment the primary model by introducing a new SPP layer and BN-ReLU-Conv block parallel to the SPP layer and BN-ReLU-Conv layer of the primary model. The output of confidence BN-ReLU-Conv block is concatenated with the segmentation maps. The confidence estimation is obtained by blending the resulting representation with a BN-ReLU-Conv block with sigmoid activation. We prevent the gradients from flowing into the segmentation maps to ensure that the segmentation head is only trained for segmentation and not for estimating confidence. The confidence map is then used to modulate the cross-entropy loss of the segmentation maps while low confidence is penalized with a hyper-parameter $\lambda$.

Our proposed discriminative OOD detector feeds the output of DB_4 to a BN-ReLU-Conv block with 2 feature maps corresponding to logits for inliers and outliers. The logits are fed to softmax and then to the cross-entropy loss which encourages the model to classify the pixels according to the respective labels. We only train the head, DB_4 and T_3 in order to speed up learning and prevent overfitting as shown in Figure 1. We assume that initial DB_3 features are expressive enough due to ILSVRC pretraining, and that it is enough to only fine-tune DB_4 for discriminating road-driving scenes from the rest of the visual world.

## 4.4 Training

We train the primary model on Cityscapes train at half resolution. The training images are normalized with Cityscapes train mean and variance. We optimize the model on entire images for 40 epochs without jittering using ADAM optimizer, a decaying learning rate starting from 4e-4 and a batch size of 5. The learning rate for the pretrained DenseNet layers is four times smaller than the learning rate for the model head. The model achieves 70.1 % mIoU on Cityscapes val and 23.6 % mIoU on WildDash val. Due to poor generalization on WildDash, we also trained a model on

the combination of Cityscapes train and WildDash val. This model reaches 70.2% mIoU on the Cityscapes validation set.

We train the augmented primary model (DeVries & Taylor, 2018) in the same way as the primary model. This model achieves 71.1 % mIoU on Cityscapes val and 25.7 % mIoU on WildDash val.

We train our discriminative models for OOD detection in the similar way as the primary model. We use three different in-distribution (ID) datasets: Cityscapes, Cityscapes + WildDash val and Vistas. WildDash val was added to Cityscapes because models trained on Cityscapes are prone to overfitting. In order to show that training on the WildDash validation subset can be avoided, we also show results of a model instance trained on the Vistas dataset (Neuhold et al., 2017) which is much more diverse than Cityscapes.

As explained earlier, we always use ILSVRC as the training dataset for the OOD class. Unfortunately, ILSVRC images are not annotated at the pixel level. We deal with this challenge by simply labeling all ILSVRC pixels as the OOD class, and all pixels from the road-driving datasets as the ID class.

In order to account for different resolutions, we resize the ILSVRC images so that the smaller side equals 512 pixels while keeping the image proportions. We form mixed batches (Kreso et al., 2018) by taking random crops of $512 \times 512$ pixels from both training datasets (ILSVRC, in-distribution) and normalizing them using the ILSVRC mean and variance. Since there is a huge disproportion in size between the ILSVRC dataset and road-driving datasets, we oversample the road-driving datasets so that the number of images becomes approximately equal. We train the models using ADAM optimizer, a decaying learning rate and a batch size of 30, until the accuracy on WildDash val reaches 100%. Similar accuracies are also observed on ILSVRC val (OOD) and Vistas val (ID) after the training.

## 4.5 EVALUATION

We evaluate how well the considered models separate OOD and ID samples on several test datasets. We compare our discriminative OOD model with the max-softmax (Hendrycks & Gimpel, 2017), ODIN (Liang et al., 2018), trained confidence (DeVries & Taylor, 2018) and the pretrained runner-up model from the ROB Challenge which was provided by its authors (Kreso et al., 2018). We quantify performance with average precision because it shows how well the method separates the OOD and ID pixels without having to look for appropriate discrimination thresholds.

We assume image-wide ID and OOD labeling (further experiments are presented in the appendix). We label all pixels in WildDash test images #0-#140 as ID, and all pixels in WildDash test images #141-#155 as OOD. We provide two AP measures. The first one evaluates results on all negative images (#141–#155). The second one ignores altered valid images (Zendel et al., 2018) (see Figure 3) which, in our view, can not be clearly distinguished from in-distribution images. Thus, we respect the original setup of the challenge, but also provide the second performance metric for a more complete illustration of the results. Note that we consider all pixels in OOD images as "positive" responses, including the pixels of Cityscapes classes (e.g. a person in a room). Conversely, we consider all pixels in ID images as "negatives", including the pixels which do not belong to any of the road driving classes (e.g. animals on the road). Such ambiguous pixels are rare and do not compromise our conclusions.

## 4.6 RESULTS ON WILDDASH TEST

Table 1 shows the results of OOD detection based on the max-softmax criterion with two instances of our primary semantic segmentation model. The two model instances were trained on i) Cityscapes train and ii) Cityscapes train + WildDash val. The evaluation was performed on the WildDash test without the five altered valid images (cf. Figure 3) (left) and on the complete WildDash test (right). Both approaches perform rather poorly, though training on WildDash val doubles the performance.

The precision-recall curve for the model trained on Cityscapes and WildDash val can be seen in the first column of Figure 2. The precision is low even for small values of recall indicating a very poor separation of ID an OOD pixels.

Table 1: Average precision performance for OOD detection by applying max-softmax to predictions of the primary model trained on two road-driving datasets.

| Training set | WD test selection | WD test complete |
|---|---|---|
| City | 10.09 | 11.91 |
| City + WD val | 17.62 | 19.29 |

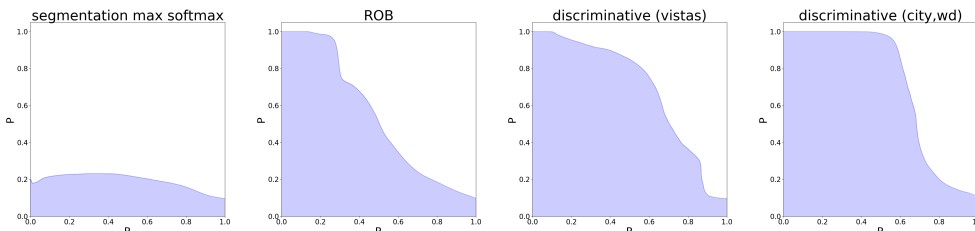

Figure 2: Precision-recall curves of OOD responses on the complete WildDash test. The columns correspond to: i) OOD detection according to max-softmax of the primary model, ii) OOD detection by recognizing foreign classes with the ROB model, iii) discriminative OOD detection trained on Vistas, and iv) discriminative OOD detection trained on Cityscapes and WildDash val.

We show the confidence that the corresponding pixel is OOD in the second column of Figures 3, 4 and 5, where red denotes higher confidence. We see that OOD responses obtained by the max-softmax approach do not correlate with epistemic uncertainty. Instead they occur on semantic borders, small objects and distant parts of the scene, that is on details which occur in most training images.

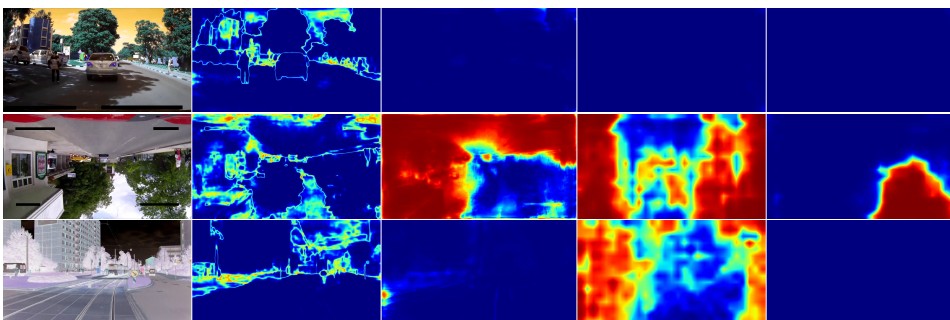

Figure 3: OOD detection in three of the five altered valid scenes from WildDash test (images #141, #148 and #151). These images were ignored in experiments labeled as "WD test selected". The columns correspond to: i) original image, ii) best result from Table 1, iii) discriminative OOD detection with the ROB model, iv) discriminative OOD detection trained on Vistas, and v) discriminative OOD detection trained on Cityscapes and WildDash val. Red denotes the confidence that the corresponding pixel is OOD which can be interpreted as epistemic uncertainty.

Table 2 shows the OOD detection with the augmented primary model and trained confidence. This approach achieves a better mIoU performance than the basic segmentation model trained only on Cityscapes. This suggests that training with uncertainty alleviates overfitting. Still, the uncertainty estimation itself proves to be the worst predictor of OOD pixels among all approaches considered in our experiments. This suggests that the confidence head is "confident" by default and must be taught to recognize the pixels which should be uncertain. Since this model performed poorly we do not show its precision-recall curve nor its OOD segmentation results.

Table 3 shows OOD detection results for a model that has seen indoor scenes (OOD) and road-driving scenes (ID) by training on all four datasets from the ROB challenge. We perform OOD detection using: max traffic softmax (maximum softmax value of ID classes) and sum traffic soft-

Table 2: Average precision for OOD detection by the augmented primary model featuring a trained confidence head.

| Training set | WD test selection | | WD test complete | |
|---|---|---|---|---|
| | max-softmax | confidence | max-softmax | confidence |
| City | 10.61 | 9.52 | 15.62 | 11.38 |

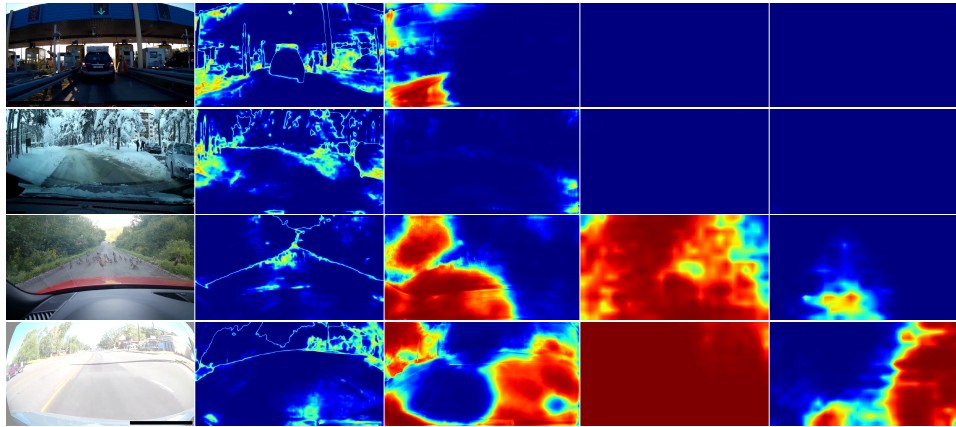

Figure 4: OOD detection in in-distribution WildDash test images. We use the same column arrangement as in Figure 3. The row three is interesting because it contains animals which are classified as ID by the model trained on WildDash val. This is likely due to the fact that WildDash val contains images with animals on the road (although not ducks). Notice finally that the model instance trained on WildDash val performs better on distorted images.

max (sum of softmax values of ID classes). Former is the OOD detection criterion from the the original paper. Latter is the total probability of ID classes. Calculating it turns the ROB model into discriminative OOD detector with ScanNet as the outlier dataset. There is a significant jump in average precision between these models and approaches trained only on road-driving scenes. Sum traffic softmax works better than max traffic softmax so we analyze it further. Interestingly, this model recognizes most pixels in the five artificially altered negative WildDash test images as ID (column 3, Figure 3). The model works well on ID images (Figure 4), however it makes some errors in some OOD images. The third column of Figure 5 shows that some pixels like ants (row 2) are recognized as ID samples. Interestingly, the model recognizes pixels at people as ID, even though they are located in an OOD context (row 3 in image 5).

We also show the precision-recall curve for this model in the second column of Figure 2. The precision is high for low values of recall. Furthermore, the precision remains high along a greater range of recall values when using probabilities for OOD detection.

Table 3: Average precision for OOD detection with the classifier trained on all four datasets from the ROB 2018 challenge: Cityscapes trainval, KITTI train, WildDash val and ScanNet train. Detection of inlier classes (sum traffic softmax) works better than using model output uncertainty for inlier classes (max traffic softmax).

| Training set | OOD approach | WD test selection | WD test complete |
|---|---|---|---|
| ROB 2018 | max traffic softmax | 65.64 | 51.29 |
| ROB 2018 | sum traffic softmax | 69.19 | 54.71 |

Table 4 shows average precision for the proposed discriminative OOD detectors which jointly train on the ILSVRC dataset and road-driving imagess from different datasets. We start with the model instance which was trained using only Cityscapes images as ID examples. Interestingly, this instance performs poorly because it classifies all WildDash test images as OOD. This result indicates that Cityscapes dataset encourages overfitting due to all images being acquired with the same cam-

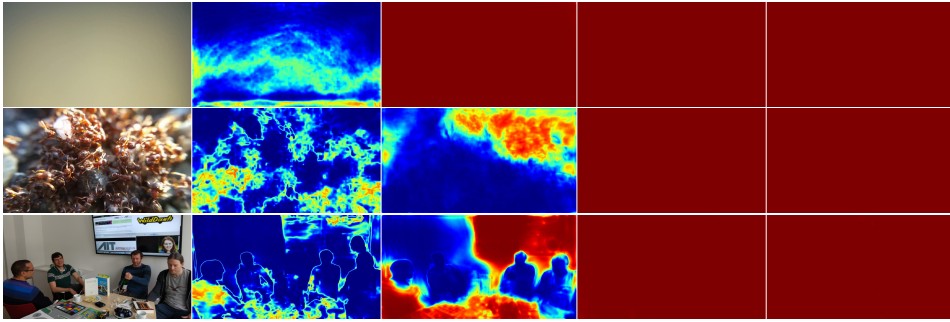

Figure 5: OOD detection in negative WildDash test images. We use the same column arrangement as in Figure 3. The ROB model classifies all non-indoor images as ID, different from the models that have seen the ILSVRC dataset.

era. The other two instances of the model perform better than all other approaches presented in this paper (including sum max softmax on ROB dataset indicating that ILSRVC is a better choice of outlier dataset than ScanNet) The model instance trained on Vistas performs significantly better than the model instance trained only on Cityscapes. Still, we obtain the best results with the instance that has seen WildDash validation set. This suggests that a model that needs to work in challenging environments also needs to see challenging examples during training. Interestingly, these two instances recognize pixels from artificially altered images as ID samples, which is evident from the drop of performance between the two columns in Table 4 as well as from columns 4 and 5 in Figure 3. Finally, these models do not recognize ID classes in OOD context: people sitting in a room are classified as OOD samples as shown in row 3 in Figure 5.

Table 4: Average precision for discriminative OOD detection on the WildDash test dataset. The OOD detection model is jointly trained on ILSVRC (OOD pixels) and road-driving images from different datasets (ID pixels).

| Training set | WD test selection | WD test complete |
|---|---|---|
| city,img | 32.11 | 24.83 |
| city,wd,img | 96.24 | 71.43 |
| vistas,img | 89.23 | 67.44 |

Precision-recall curves for the model instance trained on Vistas and the model instance trained on Cityscapes + WildDash can be seen in Figure 2, in columns 3 and 4 respectively. The curve for the model that has only seen the Vistas dataset slopes relatively constantly, while the curve for the model that has seen the WildDash validation set remains constant and high and then drops suddenly. This drop is due to altered valid images shown in Figure 3.

Finally, we show a few difficult cases in Figure 6 to discuss the space for improvement. Rather than visualizing the classification (which has sharp edges), we show confidence that the pixel is OOD. The first two rows contain images which are clearly inliers, however our discriminative models suspect that some of their parts are OOD. This is probably caused by existence of ID classes in ILSVRC images (e.g. caravan, sports car, dome). Our models are not expressive enough to indicate which ILSVRC classes caused these errors. Images in rows 3 and 4 are ID images that contain OOD objects, in this case animals. Future models would benefit by better considering the overlap between ILSVRC and the primary training dataset. Furthermore, the predictions are very coarse due to image-wide annotations of the training datasets. Finer pixel-level predictions would likely be obtained by training on images that contain both ID and OOD pixels.

## 4.7 RESULTS ON OTHER DATASETS

Table 5 shows how well the proposed OOD-detection models generalize to datasets which were not seen during training. Rows 1 and 3 show the difference between using Vistas and Cityscapes as ID dataset. When using Vistas as ID, almost no OOD pixels are detected in Cityscapes. On the

other hand, when using Cityscapes as ID, most Vistas pixels are classified as OOD. This suggests that Cityscapes poorly represents the variety of traffic scenes. Row 2 shows that almost all Pascal VOC2007 pixels are classified as OOD. This finding complements the results from Figure 5, and suggests that using ILSVRC as outlier dataset is able to generalize well to other outlier datasets.

Table 5: Pixel accuracy of discriminative OOD detection on various datasets. PASCAL* denotes PASCAL VOC 2007 trainval without Cityscapes classes (bicycle, bus, car, motorbike, person, train).

| Training set | Test set | OOD incidence |
|---|---|---|
| Vistas, ILSVRC | Cityscapes test | 0.01% |
| Vistas, ILSVRC | PASCAL* | 99.99% |
| Cityscapes, ILSVRC | Vistas val | 93.76% |

## 5 CONCLUSION

Graceful performance degradation in presence of unforeseen scenery is a crucial capability for any real-life application of computer vision. Any system for recognizing images in the wild should at least be able to detect such situations in order to avoid disasters and fear of technology.

We have considered image-wide OOD detection approaches which can be easily adapted for dense prediction in high resolution images. These approaches have delivered very low precision in our experiments because they unable to ignore the contribution of aleatoric uncertainty in the primary model output. We have therefore proposed a novel approach for recognizing the outliers as being more similar to some "background" dataset than to the training dataset of the primary model.

Our experiments have resulted in a substantial improvement of OOD detection AP performance with respect to all previous approaches which are suitable for dense prediction in high resolution images. ILSVRC appears as a reasonable background dataset candidate due to successful OOD detection in negative WildDash images that are (at least nominally) not represented in ILSVRC (white wall, two kinds of noise, anthill closeup, aquarium, etc). Nevertheless, our study emphasizes the need for more comprehensive background datasets. Future work will address employing these results as a guide for better direction of the annotation effort as well as towards further development of approaches for recognizing epistemic uncertainty in images and video.

Future work will address employing these results as a guide for better direction of the annotation effort as well as towards further development of approaches for recognizing epistemic uncertainty in images and video.

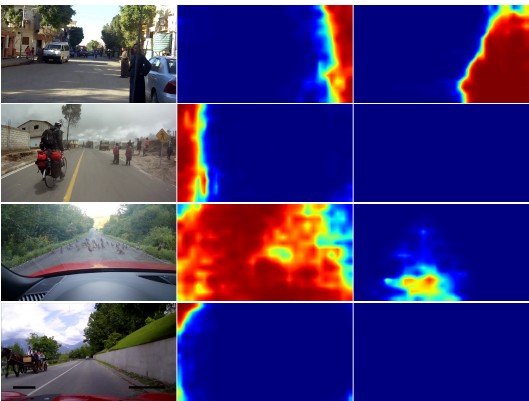

Figure 6: Examples of OOD pixel detections in positive WildDash test images. The columns correspond to: i) original image, ii) discriminative OOD detection trained on Vistas, and iii) discriminative OOD detection trained on Cityscapes and WildDash val. Red denotes the confidence that the corresponding pixel is OOD which can be interpreted as epistemic uncertainty.

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

## APPENDIX A   DENSE OOD DETECTION ON ROAD DRIVING IMAGES WITH MIXED CONTENT

The Wilddash dataset is the only publicly available dataset that provides OOD images. Unfortunately, the Wilddash OOD content is annotated only on the image level. This makes Wilddash unsuitable for testing detection of unfamiliar objects in familiar settings. We therefore propose six new datasets for that purpose.

We also propose an improved training procedure which allows the proposed discriminative OOD detection model to accurately predict borders of OOD objects. This procedure is used to train a new instance of the discriminative OOD model which is going to be evaluated in experiments.

Finally, we present and discuss experiments which compare the AP performance across several OOD models and datasets.

### A.1   TEST DATASETS

In order to be able to evaluate how different OOD detection methods perform when OOD pixels are present in ID scenes, we create six new datasets- Three of these datasets include images which contain both ID and OOD pixels. We shall use these datasets for evaluating various OOD detection approaches. The remaining three datasets are designed for control experiments in which we shall explore whether the evaluated OOD detection approaches are able to react to pasted ID content.

We obtain the first two datasets by pasting Pascal VOC 2007 animals of different sizes to images from Vistas val. Pascal was chosen because it contains many densely annotated object instances which are out-of-distribution for road-driving scenes. We do not use Vistas train at this point since we wish to preserve it for OOD training. The three control datasets are formed by pasting objects across images from road driving datasets. The final dataset contains a selection of Vistas images in which a significant number of pixels are labeled as the class 'ground animal'. The last dataset is the closest match to a real-world scenario of encountering an unexpected object while driving, however, the number of its images is rather small (hence the need for datasets obtained by pasting).

#### A.1.1   PASCAL TO VISTAS 10%

We start by locating Pascal images with segmentation groundtruth which contain any of the 7 animal classes: bird, cat, cow, dog, horse and sheep. We select 369 large Pascal objects from their original images using pixel-level segmentation groundtruth. For each selected object we choose a random image from Vistas val, resize the object to cover at least 10% image pixels and then paste the object at random image location. This results in 369 combined images. Examples of the combined images are shown in column 1 in Figure 7.

#### A.1.2   PASCAL TO VISTAS 1%

A possible issue with resizing objects before pasting might be that the OOD model may succeed to detect the pasted objects by recognizing the resizing artifacts instead of the novelty. In order to address this issue, we form another dataset as follows. We iterate over all instances of Pascal objects, we choose a random image from Vistas val and paste the object without any resizing only if it takes at least 1% image pixels. This results in 31 combined images. This datasets is more difficult than the previous one since OOD patches are much smaller. Examples can be seen in the first column of Figure 8.

#### A.1.3   CITY TO CITY

We create this dataset by pasting a random object instance Cityscapes val at a random location of a different random Cityscapes validation image. The only condition is that the object instance takes at least 0.5% of the cityscapes image. No preprocessing is performed before the pasting. Performance on this set indicates whether a model detects OOD pixels due to different imaging conditions in which the patches were acquired. This dataset contains 288 images. Examples can be seen in the first column Figure 9.

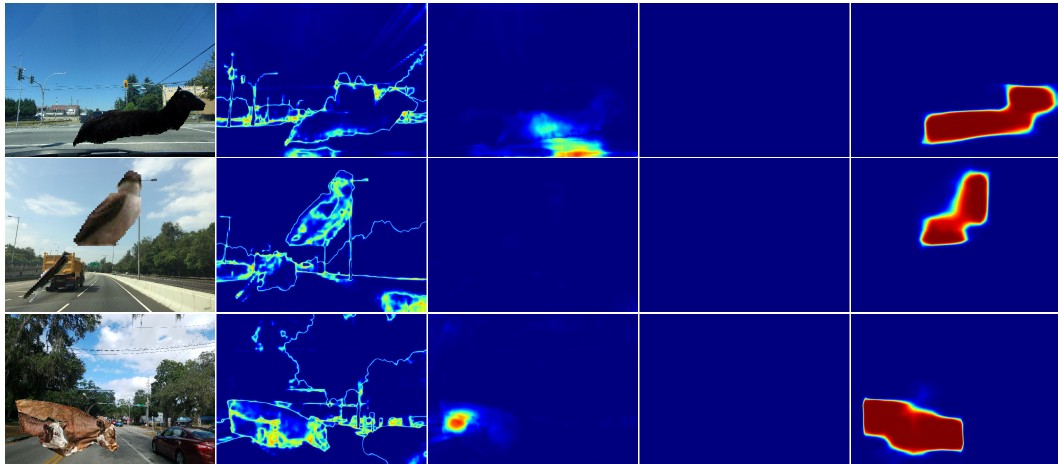

Figure 7: OOD detection in Vistas images with pasted Pascal objects that take up at least 10% of the image. The columns correspond to: i) original image, ii) max-softmax of the primary model (cf. Table 1), iii) OOD detection with the ROB model (cf. Table 3), iv) discriminative OOD detection trained on entire images from ILSVRC (OOD) and Vistas train (ID) (cf. Table 4), and v) discriminative OOD detection trained on entire ILSVRC images (OOD), and ILSVRC bounding boxes (OOD) pasted over Vistas images without ground animals (ID). Red denotes the confidence that the corresponding pixel is OOD, which can be interpreted as epistemic uncertainty. Max-softmax of the primary model detects borders. The model trained according to A.2 manages to accurately detect the OOD shape. The ROB model manages to detect the position the pasted patch, while the discriminative model trained only on the whole OOD images does not detect any of the pasted patches.

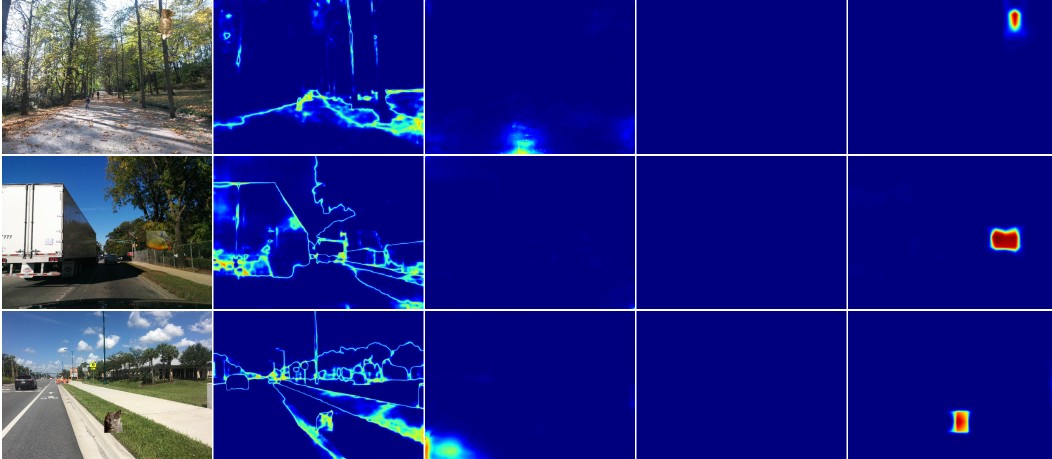

Figure 8: OOD detection in Vistas images with pasted Pascal objects that take at least 1% of the image. We use the same column arrangement and colors as in Figure 7. The model trained as described in A.2 is able to detect even the relatively small objects pasted into a similar background. The ROB model fails to detect the location of the pasted patch.

### A.1.4 VISTAS TO CITY

We create this dataset by pasting a random object instance from Vistas val into a random image from Cityscapes val. The pasted instance has to take at least 0.5% of the Cityscapes image. No preprocessing is performed before the pasting. Performance on this set indicates whether the model is able to detect different camera characteristics of the patch rather than real OOD pixels. This dataset contains 1543 images. Some examples are shown in Figure 10.

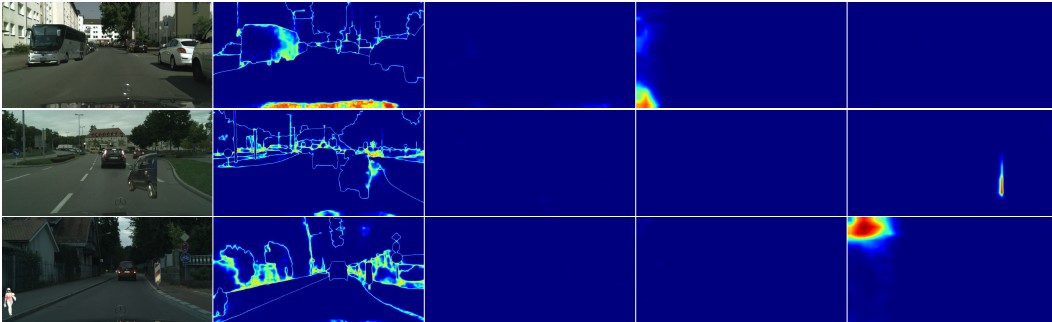

Figure 9: OOD detection in Cityscapes images with pasted Cityscapes instances that take at least 0.5% of the image. We use the same column arrangement and colors as in Figure 7. None of the models accurately detect the pasted patches. The fourth model seems to react to borders of pasted content (row 2).

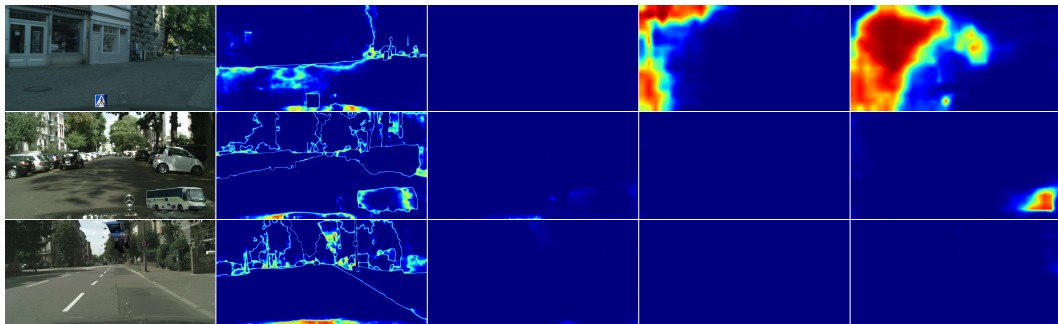

Figure 10: OOD detection in Cityscapes images with pasted Vistas instances that take at least 0.5% of the image. We use the same column arrangement and colors as in Figure 7. The fourth model experiences trouble with atypical Cityscapes images (row 1) and detects borders of the pasted patches.

### A.1.5 SELF TO SELF

We create this dataset by pasting a randomly selected object instance from a Vistas image to a random location in the same image. The object instance had to take at least 0.5% of the vistas image. No preprocessing was performed before the pasting. Performance on this set indicates whether the model is able to detect objects at unusual locations in the scene. This set contains 1873 images. Some examples can be seen in Figure 11.

### A.1.6 VISTAS ANIMALS

This dataset is a subset of Vistas training and validation images which contain instances labeled 'ground animal' that take at least 0.7% of the image. This set is closest to the real-world scenario of encountering unknown objects in ID road driving scenes. Unlike in images with pasted Pascal animals, OOD detection is unable to succeed by recognizing the pasting artifacts or different imaging conditions. This set contains 8 images. Three of those are shown in the first column of Figure 12.

### A.2 TRAINING DATASET

In order to be able to cope with images containing both ID and OOD pixels, we perform the following changes to the training dataset. First, we remove all Vistas images which contain instances of the class 'ground animal' from the training split, regardless of the instance size. We denote the vistas dataset without animals as "vistas-a". Then, we select 544 546 ILSVRC images in which bounding box annotation is available. Each of the selected ILSVRC images is used only once during training, either as i) a single image or ii) a combined image obtained by pasting the resized bounding box to a random location of a random Vistas train image. In the former case, the bounding box is labeled

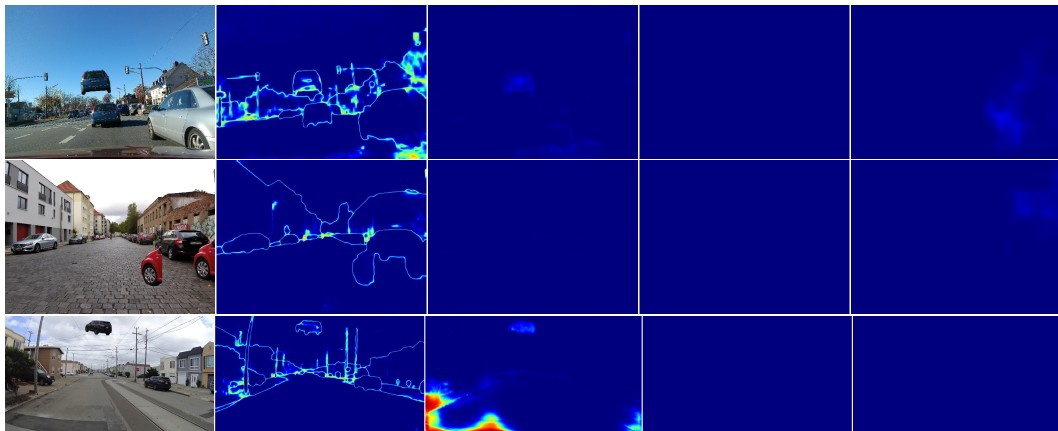

Figure 11: OOD detection in Vistas images that contain objects copied and pasted from the same image We use the same column arrangement and colors as in Figure 7. None of the models detect the pasted patches.

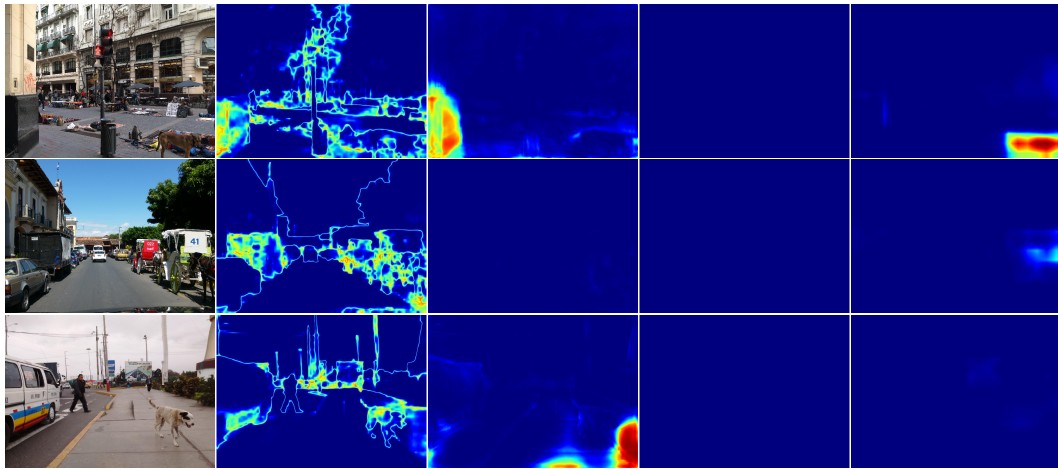

Figure 12: OOD detection in Vistas images that contain the class 'ground animal'. We use the same column arrangement and colors as in Figure 7. Only the fourth model manages to accurately segment an animal in row 1, and reacts to animals in other two images. The ROB model detects some parts as OOD however those regions do not correspond to animal locations.

as OOD while the rest of ILSVRC image is ignored. In the latter case, the bounding box is resized to contain 5% pixels of the Vistas image, the resulting ILSVRC pixels are labeled as OOD and the rest is labeled as ID. In both cases the resulting image is resized so that the shorter dimension equals 512 pixels and randomly cropped to the resolution of 512×512.

### A.3  MODEL AND TRAINING

We use the same fully convolutional discriminative OOD model as described in Section 4.3. The model is trained according to procedure described in Section 4.4, except that the training dataset is composed as described in A.2.

A.4 EXPERIMENTAL RESULTS

Tables 6 and 7 show the average precision performance for all OOD models described in the paper, together with the new instance of the discriminative model which is trained on the train dataset described in Section A.2 of this appendix. The models are evaluated on all six test datasets presented in section A.1. We compare the achieved performance with the respective results on the WildDash test dataset which we copy from Section 4.

Table 6: Average precision for discriminative OOD detection on the test datasets with images that have both ID and OOD pixels. Labels stand for max-softmax of the primary model (ms), max-softmax of the model with trained confidence (ms-conf), primary model trained for the ROB challenge (ROB), and the discriminative model (discrim). We show results for the following datasets: WildDash (wd), Cityscapes (city), the four ROB 2018 challenge datasets (ROB), full imagenet (img), the subset of imagenet with annotated bounding boxes (imb_bb), the full vistas dataset (vistas) and the vistas dataset with images containing ground animals removed (vistas-a).

| Model | Training set | PascalVistas10 | PascalVistas1 | VistasAnimals | Wilddash selection |
|---|---|---|---|---|---|
| ms | city | 28.81 | 9.91 | 6.05 | 10.09 |
| ms | city, wd | 34.27 | 8.8 | 6.79 | 17.62 |
| ms-conf | city | 26.09 | 8.04 | 5.94 | 10.61 |
| ROB | ROB | 25.65 | 4.55 | 2.96 | 69.19 |
| discrim | city, img | 34.07 | 3.19 | 2.28 | 32.11 |
| discrim | city, wd, img | 24.46 | 3.19 | 4.59 | **96.24** |
| discrim | vistas, img | 13.14 | 2.39 | 2.4 | 89.23 |
| discrim | vistas-a, img_bb | **87.87** | **78.58** | **25.61** | 68.59 |

Images 7, 8, 9, 10, 11 and 12 show the responses of OOD detection for various models. Red denotes the confidence that the corresponding pixel is OOD, which can also be interpreted as epistemic uncertainty. The columns in these images correspond to: i) original image, ii) max-softmax of the primary model (cf. Table 1), iii) OOD detection with the ROB model (cf. Table 3), iv) discriminative OOD detection trained on entire images from ILSVRC (OOD) and Vistas train (ID) (cf. Table 4), and v) discriminative OOD detection trained on entire ILSVRC images (OOD), and ILSVRC bounding boxes (OOD) pasted over Vistas images without ground animals (ID), as described in Section A.2.

These results once again show that the max-softmax approach predicts high uncertainty on object borders. Both the ROB model and the discriminative model trained on entire images fail to detect OOD patches in many images (Figures 7, 8 and 12). Poor performance of the ROB model is expected since its training datasets do not include animals. Poor performance of the discriminative model trained on entire images is also understandable since none of its training images had a border between ID and OOD regions.

The discriminative model which we train according to Section A.2 delivers the best overall performance. It is able to detects OOD patches even on very small pasted objects (cf. Figure 8) and genuine animals in Vistas images (cf. Figure 12). We note that this model occasionally detects borders of ID patches (row 2 in Figure 9 and row 2 in Figure 10) which suggests that results on PascalToVistas may be a little too optimistic. We also note that this model sometimes misclassifies parts of Cityscapes images.

Genuine Vistas images with ground animals (VistasAnimals) is the most difficult dataset for all models, however the discriminative model trained according to Section A.2 clearly achieves the best performance (cf. Figure 12, row 1).

Table 7 shows average precision for pasted content detection on the three control datasets. The AP on control datasets indicates if the model is able to distinguish between the ID image and the ID pasted region. High AP on these datasets means that the corresponding OOD model detects differences in imaging conditions or unexpected object locations between the ID image and the pasted ID patch. High performance on control datasets would indicate that success on PascalToVistas datasets is the result of detecting the process of pasting instead of the novelty of the Pascal classes. The score of the best discriminative model indeed indicates that part of its success on PascalToVistas datasets comes from recognizing pasting interventions.

Table 7: AP for detection of pasted content in the three control datasets. Labels stand for max-softmax of the primary model (ms), max-softmax of the model with trained confidence (ms-conf), primary model trained for the ROB challenge (ROB), and the discriminative model (discrim). We show results for the following datasets: WildDash (wd), Cityscapes (city), the four ROB 2018 challenge datasets (ROB), full imagenet (img), the subset of imagenet with annotated bounding boxes (imb_bb), the full vistas dataset (vistas) and the vistas dataset with images containing ground animals removed (vistas-a).

| Model | Training set | CityCity | VistasCity | Self |
|---|---|---|---|---|
| ms | city | 3.96 | 7.42 | 5.74 |
| ms | city, wd | 3.49 | 6.81 | 7.42 |
| ms-conf | city | 3.61 | 6.69 | 5.34 |
| ROB | ROB | 4.64 | 13.37 | 5.95 |
| discrim | city, img | 2.15 | 45.48 | 2.92 |
| discrim | city, wd, img | 2.68 | 42.82 | 3.21 |
| discrim | vistas, img | 2.39 | 9.14 | 3.56 |
| discrim | vistas-a, img_bb | 7.62 | 34.12 | 19.74 |

## A.5 CONCLUSION

Experiments show that training on ILSVRC bounding boxes pasted above Vistas images is able to deliver fair open-set dense-prediction performance. In particular, our model succeeds to detect animals in road-driving images although it was not specifically trained for that task, while outperforming all previous approaches by a wide margin. We believe that these results strengthen the conclusions from the main article and provide useful insights for future work in estimating epistemic uncertainty in images.

## APPENDIX B    DENSE OOD DETECTION ON UCSD PED2 DATASET

We further explore performance of the proposed dense OOD-detection approach on UCSD Ped2 dataset which contains video sequences of pedestrian scenes. The test subset contains 12 video clips in which each image is annotated with pixel-level binary masks which identify regions with anomalies. The train subset contains no anomalies. It is similar to Vistas animals dataset (subsection A.1.6) in that it contains images that are a true mix of anomalous and non-anomalous regions. However, the problem of OOD detection is not the same as UCSD anomaly detection since UCSD anomalies are connected to motion. For example, a person walking next to a bike is not an anomaly. On the other hand, a person riding a bike and a person walking on the grass are both considered anomalous. Still, it is interesting to see how discriminative OOD detector behaves on a dataset quite different to previously described road driving datasets. Furthermore, since there is overlap between UCSD anomalies and OOD detection, AP can be obtained and used to get a sense of the behavior of the detector.

### B.1    UCSD PED2 DATASET

The UCSD dataset contains grayscale video of pedestrian walkways taken with a stationary camera. We focus on video clips from the Ped2 subset which contains 16 training and 12 testing video clips. All these clips have been acquired from the same viewpoint and by the same camera, so that the pedestrian movement is parallel to the image plane. Ped2 test subset contains anomalies which are not present in Ped2 train; these anomalies correspond to bikers, skaters and small carts. Examples of images from UCSD Ped2 dataset are shown in the first column of Figure 13.

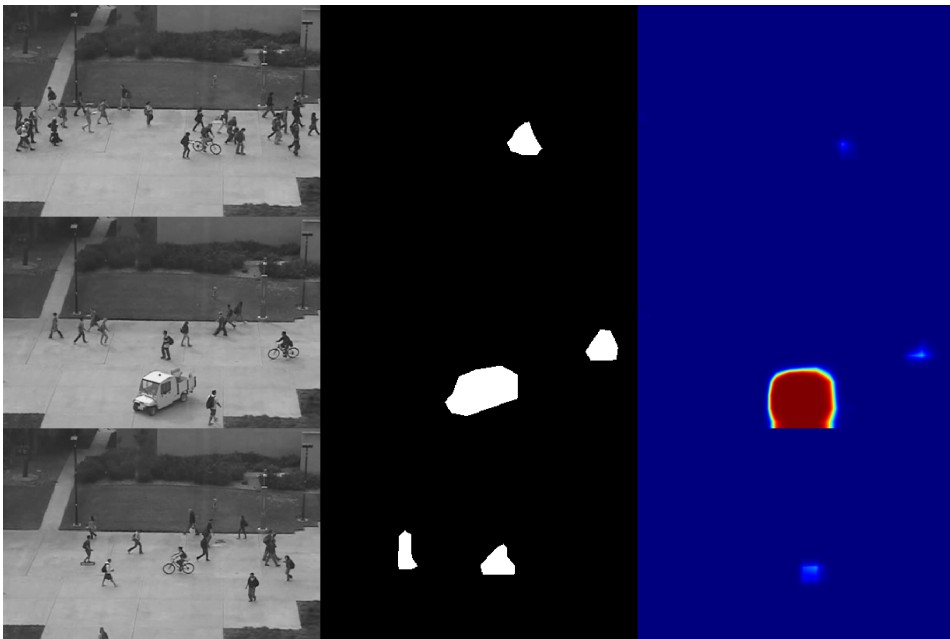

Figure 13: OOD detection in UCSD Ped2 dataset. The first column contains original images, the second ground truth annotations, and the third OOD discriminator output. Red denotes the confidence that the corresponding pixel is OOD. The model easily detects the cart as OOD. It detects the wheels of the bicycle as OOD, but bike riders are detected as ID. The skater is also detected as ID.

### B.2    TRAINING DATASET

The training dataset is built as described in in A.2 except that we use images from UCSD Ped2 training dataset in place of Vistas images.

### B.3 MODEL AND TRAINING

We use the same setup for training on the UCSD as described in A.3, but we also randomly convert half of the training images into grayscale images.

### B.4 EXPERIMENTAL RESULTS

Figure 13 shows the response of the discriminative model trained on the UCSD Ped2 dataset. Table 8 shows the result of the discriminative model on the complete UCSD Ped 2 dataset, as well as on three of its test sequences: 1, 4, and 8. Sequence 1 contains a cyclist and a relatively large number of pedestrians, sequence 4 contains a small cart, while sequence 8 contains a cyclist and a skater but less pedestrians. The AP is highest on sequence 4 where the motion anomaly occurs on an object which is clearly OOD. We report lower AP on sequences with cyclists and skaters. There are two causes for this deterioration. Firstly, our model is unable to detect skaters as OOD due to their similarity with pedestrians. Secondly, cyclists (together with their bikes) are labeled as anomalies in the ground truth annotations. Bikes are not necessarily labeled as anomalies if they are not being ridden. Our model recognizes majority of the pixels of bikes as OOD (ridden or not), while riders themselves are recognized as ID, again due to similarity with pedestrians. This discrepancy considerably decreases our AP.

Table 8: AP for OOD detection on the whole UCSD Ped2 test dataset, as well as on sequences 1, 4 and 8 from that dataset, denoted as S1, S4 and S8 respectively.

| Model | Training set | UCSD Ped2 | UCSD Ped2 S1 | UCSD Ped2 S4 | UCSD Ped2 S8 |
|-------|--------------|-----------|--------------|--------------|--------------|
| discrim | UCSD, img_bb | 48.49 | 37.08 | 83.60 | 40.24 |

### B.5 CONCLUSION

Experiments show that unexpected objects can be detected in pedestrian walkways by training on ILSVRC bounding boxes pasted above UCSD images. This further supports the conclusion that ILSVRC is a good choice as an outlier distribution for a wide variety of training datasets.

