# OpenReview forum: "Discriminative out-of-distribution detection for semantic segmentation"
_ICLR.cc/2019/Conference_

### Official Review · AnonReviewer1 · 2018-10-29
**Difficult to understand**

**Rating:** 3
**Confidence:** 3

**Review:**

ML models are trained on a predefined dataset formed by a set of classes. Those classes use to be the same ones for training and testing. However, what happen when during testing time images with classes unseen during training are shown to the model? This article focus in this problem which is not currently taking much attention by the mainstream research community and is of great importance for the real world applications.

This article tries to detect areas of the image where those out-of-distribution situations appear in semantic segmentation applications. The approach used is by training a classifier that detects which pixels are out of distribution. For training two datasets are used: the dataset of interest and another different one. The classifier learns to detect if a pixel is from the dataset of interest or from another distribution.

The main problem I found with this article is that I couldn't fully understand it. Maybe because the text needs a bit more of review and improvement or maybe because Im not very familiar with the topic. Moreover the article is 10 pages while it is encouraged to be 8. I find that the method of the paper is quite simple and can be explained more straight forward and in less pages. The related work section overlaps a lot with the intro, I suggest to combine both. First two paragraphs of the method seam that should be in the intro. Model details from the experiments I consider that should be explained in the method. I miss a figure explaining the architecture of the model. Why using the semantic segmentation model proposed and no something standard? For instance Tiramisu (That is also based on dense layers). Note that the method used for semantic segmentation is 10 points lower than the SOTA in Cityscapes. Figure 1 is impossible to read as the captions are too small. The representations of figures 2-5 are difficult to interpret. There is no comparison to SOTA

---

> ### Author Response · Authors · 2018-11-08
> **Reply to AnonReviewer1**
>
> Thank you for your review. We answer your concerns as follows.
>
> > The main problem I found with this article is that I couldn't fully understand it.
> > Maybe because the text needs a bit more of review and improvement
> > or maybe because Im not very familiar with the topic.
>
> We are sorry the paper was not clear to you. Please provide more specific information regarding which parts of the paper could be clarified.
>
> > Moreover the article is 10 pages while it is encouraged to be 8.
> > I find that the method of the paper is quite simple and can be explained
> > more straight forward and in less pages.
>
> Current method section (section 3) is less than one page long. It appears to us that shortening that section would not decrease the number of pages. We could move figures 2-5 to the appendix although we feel that leaving them as they are would result in a better flow.
>
> > The related work section overlaps a lot with the intro, I suggest to combine both.
>
> We shall resolve some redundancies which we introduced for the convenience of the reader.
>
>     remove the second-to-last paragraph and shorten the last paragraph in the introduction
>     remove the last paragraph in the related work
>
> > First two paragraphs of the method seam that should be in the intro.
> > Model details from the experiments I consider that should be explained in the method.
> > I miss a figure explaining the architecture of the model.
>
> We shall refactor and shorten the first two paragraphs of section 3 and add the figure.
>
> > Why using the semantic segmentation model proposed and no something standard?
> > For instance Tiramisu (That is also based on dense layers).
> > Note that the method used for semantic segmentation is 10 points lower than the SOTA in Cityscapes.
>
> As you noted in your review, OOD detection on pixel level has not been previously investigated. We prefer to focus on baseline models at this stage in order to simplify conclusions as well as speed-up the training.
>
> SOTA on cityscapes achieve high benchmark results by recovering fine spatial detail lost due to downsampling. Our segmentation models are not as accurate on object borders or small objects like poles, but work reasonably well. When it comes to OOD detection, we are more interested in existence of OOD regions, rather than their exact outlines. Furthermore, as table 5 shows, cityscapes is in many ways a very specific dataset (single camera, nice weather conditions, German cities). Consequently, chasing SOTA on cityscapes is likely to poorly affect max-softmax OOD detection due to overfitting. Using simpler models is also a way of regularization.
>
> Table 3 in our original submission includes OOD-detection results obtained with the model which achieves 77.1 mIoU on Cityscapes. The table shows only results of OOD-detection by classification into foreign classes since this approach worked much better than max-softmax. In the revised paper we show the max-softmax results as well.
>
> We agree that densely connected layers are a very good choice for semantic segmentation. However, tiramisu would not be a suitable choice for our experiments due to following reasons:
>
>     it has a thick up-sampling path which complicates training on large images due to large memory requirements
>     the Tiramisu paper proposes exotic downsampling paths for which there are no ImageNet-pretrained parameters available; consequently this would require training from scratch and lead to at least a 10-fold increase in training time and loss of accuracy due to overfitting.
>
> > Figure 1 is impossible to read as the captions are too small.
>
> We shall improve the captions in the revised paper.
>
> > The representations of figures 2-5 are difficult to interpret.
>
> Could you please be more specific about what could be done to clarify these figures?
>
> > There is no comparison to SOTA
>
> To the best of our knowledge, there is almost no previous work in OOD detection on pixel level. Previous work in OOD detection focuses on classification tasks on entire images. We adapt these approaches for dense OOD-detection and show that our approach performs better. Kendall and Gal (2017) model epistemic uncertainty on pixel level, although they do not use it for OOD detection. Our early experiments with this approach resulted in poor OOD detection performance. WildDash is the first semantic segmentation benchmark that introduces OOD images. Most of existing submissions on WildDash come without an accompanying paper, so it is not clear what, if anything, was used for OOD detection.

---

### Official Review · AnonReviewer3 · 2018-10-30
**Interesting results, good direction to follow**

**Rating:** 7
**Confidence:** 5

**Review:**

This paper aims to detect out-of-distribution pixels for semantic segmentation, which is a good direction for researchers in this field to explore. As the authors point out, recent semantic segmentation systems surpass 80% mIoU on Pascal VOC 2012  and  Cityscapes, which is a good achievement. Unfortunately, most existing semantic segmentation datasets assume closed-world evaluation which means that they require predictions over a predetermined set of visual classes. This work utilize data from other domain to detect undetermined classes, thus can model uncertainty better in an explicit way. I just have minor comments.

1. When you perform training, do you train from scratch or from a pre-trained model? If using pre-trained model, then ILSVRC is not actually pure OOD pixels.

2. How to interpret the results in Table 5?

---

> ### Author Response · Authors · 2018-11-08
> **Reply to AnonReviewer3**
>
> Thank you for your review! We answer your questions as follows.
>
> > When you perform training, do you train from scratch or from a pre-trained model?
>
> Parameters of the feature extractor were initialized from ImageNet pre-trained models. All heads are trained from scratch.
>
> > If using pre-trained model, then ILSVRC is not actually pure OOD pixels.
>
> We do not perceive that as a problem neither in discriminative OOD detection (where we train on road driving vs ILSVRC) nor in single-class OOD detection (where we train on road driving images and rely on max-softmax). In discriminative OOD, we cast the problem as binary classification where pre-training can only help. In single-class OOD (max-softmax), the classifier is fine-tuned through 40 epochs on a road driving dataset. Previous work on catastrophic forgetting suggests this likely results in a complete oblivion of features for ILSVRC classes which are not seen in road driving datasets.
>
> Please note that we also successfully detect OOD pixels in WildDash negative images that (at least nominally) do not have anything to do with ILSVRC (white wall, two kinds of noise, anthill closeup, aquarium, etc).
>
> Maybe we do not understand your concerns. Could you please clarify?
>
> > How to interpret the results in Table 5?
>
> Table 5 shows how well the proposed OOD-detection models generalize to datasets which were not seen during training.
>
> Rows 1 and 3 show the difference between using Vistas and Cityscapes as ID dataset. When using Vistas as ID, almost no OOD pixels are detected in Cityscapes. On the other hand, when using Cityscapes as ID, most Vistas pixels are classified as OOD. This suggests that Cityscapes poorly represents the variety of traffic scenes.
>
> Row 2 shows that almost all Pascal VOC 2007 pixels are classified as OOD. This finding complements the results from figure 4, and suggests that training OOD detection on ILSVRC is able to generalize to other datasets.

---

### Official Review · AnonReviewer2 · 2018-11-07
**Discriminative out-of-distribution detection for semantic segmentation**

**Rating:** 4
**Confidence:** 4

**Review:**

Summary:
This paper addresses the problem of out-of-distribution detection for helping the segmentation process. Therefore, the detection is performed on a pixel basis. The application of the approach is to datasets used for autonomous driving, where semantic segmentation of the view of the road is a typical application. Since in a road view there will be pixels that are projections of objects that are likely not in the set of classes known by the semantic segmentation algorithm, it makes sense to flag them as being out of distribution (OOD), or not known, or to assign to them a low confidence level. The proposed approach is trivial: train a binary classifier that distinguishes image patches from a known set of classes from image patches coming from an unknown (background class). The classifier output applied at every pixel will give the confidence value. While there are different dataset options to represent the known classes, the background class is represented by images from ILSVRC. The results show that for the segmentation application the approach works better than using an adaptation of more elaborate out-of-distribution methods.

Quality and clarity:
The paper is well organized and is described very clearly and provides an ok set of results, despite the simplicity of the approach.

Originality and significance:
Unfortunately, I do not see any relevant technical novelty, and this is a major issue. Perhaps the only significant conclusion about this paper is that before designing a new OOD detector, if representing the set of “unknown” classes with ILVRC is reasonable, then it makes sense to simply train a binary classifier and see how it works.

Besides the novelty, I disagree with the way the paper has been positioned and motivated. It brings into play epistemic and aleatoric uncertainty concepts to justify (the simplicity of) the approach, and it overlooks a large body of machine learning (novelty detection, one-class classification, …). This is also a major issue.


Additional comments:

One of the biggest motivations for this work is that other approaches do not distinguish between epistemic and aleatoric uncertainty and this is why they do not work. This is regarded as a distinctive advantage of the proposed approach. It is claimed that the proposed formulation is insensitive to any aleatoric uncertainty. On the other hand, the paper is written in a way that ignores a large body of literature that goes under the name of “novelty detection”, “anomaly detection”, “one-class classification”, and related names. So, I am wondering how the approaches just mentioned compare with the proposed method, when epistemic and aleatoric uncertainty become part of the discussion. Isn’t every novelty detector insensitive to aleatoric uncertainty as well? Could the Authors clarify what they claim with that statement, while considering a broader view?

The paper should relate to the literature mentioned above. In particular, I would point the Authors to a couple of recent works that seem to precisely contradict the premises of the proposed approach, which are given at the beginning of section 3:

- Adversarially Learned One-Class Classifier for Novelty Detection, CVPR 2018
- Generative Probabilistic Novelty Detection with Adversarial Autoencoders, arXiv, July 2018.


Again, related to novelty detection, it looks like the proposed approach still requires tuning one or more thresholds. Therefore, it would not be that different from tuning the threshold of a novelty detector, or a one-class classifier. It would have strengthened the paper if the approach was compared also to a novelty detector.

It is not clear if the fully convolutional OOD detector is working on a patch or on the entire image. If it is a patch, of what size?

Page 4, define the “ID” acronym.

---

> ### Author Response · Authors · 2018-11-08
> **Reply to AnonReviewer2, Part 1**
>
> Thank you for your review! We answer your questions as follows.
>
> > Unfortunately, I do not see any relevant technical novelty, and this is a major issue.
>
> In our opinion, a simple solution to a difficult problem is preferred to a complex one.
>
> > Perhaps the only significant conclusion about this paper is that
> > before designing a new OOD detector, if representing
> > the set of “unknown” classes with ILVRC is reasonable,
> > then it makes sense to simply train a binary classifier and see how it works.
>
> Our experiments suggest that representing outliers with ILSVRC might be reasonable more often than not, since our method correctly classified negative WildDash images such as a white wall, two kinds of noise, anthill closeup, aquarium, etc.
>
> A concurrent submission to ICLR 2019 shows that representing outliers with ImageNet and 80 million images is a reasonable choice for a wide selection of datasets. We feel that our submission nicely complements their work by presenting a similar idea in the dense prediction context.
>
> https://openreview.net/forum?id=HyxCxhRcY7
>
> Yes, our main conclusion is that ImageNet seems to be diverse enough to support discriminative OOD detection in diverse traffic images. The other important finding is that entropy-based OOD detection approaches (e.g. max-softmax) are not appropriate for dense prediction due to inherent aleatoric uncertainty involved.
>
> We thank for the comment, we shall revise the conclusion accordingly.
>
> > Besides the novelty, I disagree with the way the paper has been positioned and motivated.
> > It brings into play epistemic and aleatoric uncertainty concepts to justify (the simplicity of) the approach,
> > and it overlooks a large body of machine learning (novelty detection, one-class classification, …).
> > This is also a major issue.
>
> We avoided several methods based on complex GAN designs due to large space and time complexities which would cause GPU memory exhaustion and long training times. One of our in-distribution datasets contains 18000 complex road driving images (Vistas, we reduce resolution to 512x1024 pixels). This means 150 times more pixels than in CIFAR and more than 50 times more pixels than in UCSD ped2, even if we neglect diversity of scenes (different continents, night, rain, snow, ...) and cameras. We do not say it is impossible to adapt some GAN approach to OOD detection on Vistas, however that is certainly not straightforward (as we try to describe below) and therefore out of our scope.
>
> The diversity of our in-distribution dataset restricts our options since now, when we train only the discriminator, our training procedure takes an entire day. We reckon that training a GAN variant with suitable capacity would require much more time since the generator would initially produce useless counterexamples.
>
> Our memory requirements are even more restrictive since our ambition is to attach the discriminator as the second head to a regular semantic segmentation pipeline (we tried that already, it works). State of the art semantic segmentation pipelines are designed to use the whole GPU memory so that joint approaches with large generators would not be feasible on current state of the art GPUs.
>
> We thank for the comment and apologize for not discussing these issues in our original submission! We shall include these considerations and cite the corresponding previous work in the revised paper.
>
> > Isn’t every novelty detector insensitive to aleatoric uncertainty as well?
> > Could the Authors clarify what they claim with that statement, while considering a broader view?
>
> It is true that novelty detectors such as the one which you propose below would be insensitive to aleatoric uncertainty as well. However, those papers appear to address much simpler problems. For instance, the UCSD dataset involves a fixed camera and very simple scenery. Their method does not appear suitable for the diversity of the Vistas dataset.
>
> We thank for the comment! We shall include a more accurate discussion in the revised paper.
>
> > I would point the Authors to a couple of recent works that seem to precisely contradict
> > the premises of the proposed approach, which are given at the beginning of section 3
>
> An interesting concurrent submission shows experiments in which a state of the art generative model trained on CIFAR assigns larger probabilities to SVHN images than to CIFAR images. That supports premises at the beginning of section 3, that learning the probability density of real datasets is a hard and not completely solved problem (even for datasets which are much less diverse than Vistas).
>
> https://openreview.net/forum?id=H1xwNhCcYm

---

> ### Author Response · Authors · 2018-11-08
> **Reply to AnonReviewer2, Part 2**
>
>
> > Adversarially Learned One-Class Classifier for Novelty Detection, CVPR 2018
>
> We have located and inspected the code at https://github.com/khalooei/ALOCC-CVPR2018, however it appears to be out of sync with the paper: the refinement loss in the code (grep g_r_loss) does not seem to match the equation (4) in the paper. The code does not include information to reproduce the numbers from the tables. Straight-forward evaluation of the provided trained model on few UCSD images does not appear to separate inliers from outliers.
>
> > It would have strengthened the paper if the approach was compared also to a novelty detector.
>
> Thank you for your suggestion! We agree, we shall present such discussion in the revised paper.
>
> > It is not clear if the fully convolutional OOD detector is working
> > on a patch or on the entire image. If it is a patch, of what size?
>
> Our fully convolutional OOD detector operates on entire images. It outputs a dense prediction in the form of a H/32xW/32 matrix, where HxW are dimensions of the original image (cf. [long15cvpr]). One could say that our detector operates as if it were applied to RxR patches situated 32 pixels apart where R is effective receptive field of the discriminator (for DenseNet 121 finetuned on full Cityscapes images, R is around 600 pixels).
>
> > Page 4, define the “ID” acronym.
>
> ID stands for "in-distribution". We shall clarify that, thanks!

---

### Author Response · Authors · 2018-11-20
**Revision 2 of the paper**

We have revised the paper according to the reviewers' comments.

Here is the summary of changes:

1. Introduction
paragraph 3 - shorten uncertainty definitions (Reviewer 1: "text needs a bit more improvement")
paragraph 4 - better explain the need to differentiate between the two types of uncertainties (Reviewer 2: "brings into play epistemic and aleatoric uncertainity concepts to justify the simplicity of the approach")
paragraph 5 - introduce GAN-based OOD detection approaches (Reviewer 2: "it overlooks a large body of machine learning...")
paragraph 6 - clarify advantages of the proposed approach (Reviewer 2: "I do not see any technical novelty")

2. Related work
Paragraph 1 - mention anomaly and novelty detection as related fields (Reviewer 2)
Paragraph 2 - distilled and moved to introduction paragraph 4 (Reviewer 1: "related work section overlaps a lot with the intro")
Paragraph 5 - discuss generative models for OOD detection in more depth (Reviewer 2: "It would have strengthened the paper if the approach was compared also to novelty detector")
Paragraph 6 - rephrased and improved old paragraph 5 (Reviewer 1: "related work section overlaps a lot with the intro")
Paragraph 7 - removed (Reviewer 1: "related work section overlaps a lot with the intro")

3. The proposed discriminative OOD detection approach
Paragraph 1 - replaced with a short method description (Reviewer 1: "the first two paragraphs of the method seam that should be in the intro")
Paragraph 2 - shortened (Reviewer 1: "the method of the paper can e explained more straight forward")
Add Figure 1 - (Reviewer 1: "I miss a figure explaining the architecture of the model", Reviewer 2: "it is not clear if the the OOD detector is working on a patch or on the entire image")
Paragraph 3 - rephrase 1st sentence, define ID (Reviewer 2: "define the ID acronym")
Add paragraph 5 - justify model architecture and use of ILSRVC (Reviewer 2: "if representing the set of unknown classes with ILSVRC is reasonable...", Reviewer 3: "...if using pretrained model, then ILSVRC is not actually pure OOD pixels")

4.2. Datasets
Paragraph 1 - combine Vistas and Cityscapes into same paragraph (Reviewer 1: "...can be explained in less pages")

Figure 2 (old Figure 1) - enlarge the font (Reviewer 1: "Figure 1 is impossible to read as the captions are too small")

4.6. Results on WildDash test
Paragraph 5 - add max-traffic-softmax to table 3 and compare max-traffic-softmax with sum-traffic-softmax on the ROB method which achieves 77% MIoU on Cityscapes (Reviewer 1: "...method used for semantic segmentation is 10 points lower than the SOTA...")

4.7. Results on other datasets
Clearer interpretation of experimental results (Reviewer 3: "How to interpret the results in Table 5")

5. Conclusion
Paragraphs 2,3 - clarified, shortened (Reviewer 1: "explain more straight forward", Reviewer 2: "no technical novelty", Reviewer 2: "if ILSVRC is reasonable")

Added appendix B
We present results on UCSD - to compare with image-wide approach mentioned by Reviewer 2 (Adversarially Learned One-Class Classifier for Novelty Detection).

---

### Meta-Review · Area_Chair1 · 2018-12-12
**Limited novelty**

**Confidence:** 5
**Recommendation:** Reject

**Metareview:**

The paper addresses the problem of out-of-distribution detection for helping the segmentation process.

The reviewers and AC note the critical limitation of novelty of this paper to meet the high standard of ICLR. AC also thinks the authors should avoid using explicit OOD datasets (e.g., ILVRC) due to the nature of this problem. Otherwise, this is a toy binary classification problem.

AC thinks the proposed method has potential and is interesting, but decided that the authors need more works to publish.